# Impact of the COVID-19 Pandemic on Life Expectancy in South Korea, 2019–2022

**DOI:** 10.3390/healthcare13030258

**Published:** 2025-01-28

**Authors:** Soojin Song, Daroh Lim

**Affiliations:** 1Department of Health Administration, College of Nursing and Health, Kongju National University, Gongju 32588, Republic of Korea; moon5@kongju.ac.kr; 2Division of Zoonotic and Vector Borne Disease Research, National Institute of Health, Cheongju 28159, Republic of Korea

**Keywords:** COVID-19, life expectancy, top 10 causes of death, mortality

## Abstract

Objective: This study investigated changes in life expectancy due to the COVID-19 pandemic by analyzing the contributions of age, sex, and cause of death in 2019 and 2022. Methods: Korea’s simplified life table and cause-of-death statistics from 2019 to 2022 were used to assess mortality changes by age, sex, and cause of death during the pandemic. Joinpoint regression analysis was applied to detect trends, and the Arriaga decomposition method was used to quantify the contributions of age, sex, and cause of death to life expectancy changes. Results: Joinpoint regression identified a slow increase in life expectancy in 2007 and a decline in 2020, coinciding with the COVID-19 pandemic. Life expectancy decreased markedly for men (−0.36 years per year, 95%CI: −0.68 to −0.03) and women (−0.45 years per year, 95%CI: −0.71 to −0.18). Age-specific contributions revealed declines across age groups, with the steepest reductions in the older population (80 years or older: −0.35 years for men; −0.52 years for women). Women (−0.68 years) contributed more to the decline in life expectancy than men (−0.41 years). COVID-19 ranked as the third leading cause of death in 2022, significantly contributing to the decline in life expectancy among the older population (aged 80 years or older: −0.306 years for men, −0.408 years for women). Women in Korea were more affected than men, reducing the sex-specific gap in life expectancy by 0.3 years. Conclusions: The COVID-19 pandemic significantly impacted the life expectancy in Korea, particularly among older adults, with women experiencing a greater decline than men. These findings emphasize the need for targeted public health strategies to address age and sex disparities in future pandemics. Before the pandemic, non-communicable diseases such as malignant neoplasms, heart disease, and cerebrovascular disease dominated Korea’s top 10 causes of death. During the pandemic, however, COVID-19 rose to third place by 2022. Notably, intentional self-harm (suicide) contributed to an increase in life expectancy, suggesting shifts in the relative impact of various causes of death.

## 1. Introduction

Life expectancy is the average number of years a person born at age zero can be expected to live. It is a comprehensive indicator for evaluating the health status of a country or community and an essential measure for analyzing changes in mortality. Most countries experienced significant increases in life expectancy during the latter half of the 20th century [1,2], and Korea has become one of the countries with the highest life expectancy. Between 1970 and 2019, the life expectancy in Korea increased by 21.1 years, from 62.2 years to 83.3 years [3].

However, the COVID-19 pandemic caused a marked mortality shock worldwide. As of 1 July 2024, the cumulative global number of deaths per million people reached 883,896, the largest mortality rate shock since World War II. Consequently, global life expectancy decreased by 1.8 years from 72.8 years in 2019 to 71.0 years in 2021 [2,4,5]. Among European countries, Bulgaria, Slovakia, Poland, Lithuania, Hungary, Estonia, Czech Republic, and Croatia all witnessed a decline in life expectancy of more than 1.75 years between 2019 and 2021, with Bulgaria in particular experiencing the largest loss among the driving European countries, with a decline of 3.58 years. In the United States, life expectancy decreased by 2.35 years from 2019 to 2021 [2].

In Asia, life expectancy decreased by an average of 1.66 years, which is slightly lower than the global average of 1.74 years. South Asia experienced the highest loss in life expectancy (3.01 years), while East Asia showed only minimal changes [6]. In Japan, life expectancy increased slightly by 0.24 years in 2020 compared to 2019; however, it increased and then decreased by 0.15 years in 2021. Despite the pandemic’s negative effects, its impact on life expectancy was relatively minor [7].

South Korea maintained relatively low mortality rates throughout the pandemic. Cumulative deaths due to COVID-19 were 15.59 per million people in 2020, 102.28 per million in 2021, 623.38 per million in 2022, and 693.50 per million in 2023 (as of 31 December 2023) [5]. Korea was one of the first countries affected by the pandemic. It managed to minimize mortality through an effective preemptive testing, tracing, and treatment (3T) strategy. This approach was similar to that of countries, such as Denmark, Iceland, and New Zealand, where life expectancy also remained relatively stable during the pandemic [8,9].

Nevertheless, incomplete death registration systems and differences in the definitions of COVID-19-related deaths between countries complicate the accurate identification of mortality due to the indirect effects of the pandemic [6,10]. According to the Joint Technical Advisory Group on COVID Mortality Assessment of the World Health Organization (WHO) and the United Nations Department of Economic and Social Affairs (UN DESA), excess deaths include those directly attributable to the COVID-19 virus as well as indirect deaths caused by limited access to healthcare and reduced mortality rates from factors such as seasonal influenza, traffic accidents, or occupational injuries during the pandemic [6,10,11].

Numerous studies have documented changes in life expectancy due to the COVID-19 pandemic [12,13,14,15,16,17,18,19,20,21,22,23], demonstrating its significant impact worldwide. In Korea, the pandemic also negatively impacted life expectancy. Although the upward trend in life expectancy was not significantly affected in 2020 and 2021, a decline was observed in 2022, coinciding with the emergence of the Omicron variant and the downgrading of the COVID-19 crisis level from “serious” to “level 2 infectious disease”.

This study aimed to examine the changes in life expectancy due to the COVID-19 pandemic. Unlike previous studies, this study used the Joinpoint Regression Program to logically approach the changes in life expectancy before (2019) and after (2022) the COVID-19 pandemic. Subsequently, the contributions of age, sex, and cause of death were compared and analyzed using the Arriaga decomposition method.

## 2. Materials and Methods

### 2.1. Demographic and Epidemiological Datasets

This study investigated the age, sex, and causes of death in Korea during the COVID-19 pandemic. To determine the extent to which changes in mortality contributed to changes in life expectancy, we used Korea’s simplified life table and cause-of-death statistics from 2019 to 2022, provided by the Korea Statistical Information Service (KOSIS) of Statistics Korea (2023). Statistics Korea, the agency responsible for the national statistics, ensures the completeness and reliability of KOSIS.

Considering the importance of Korea’s national health priorities, including deaths from COVID-19 and age- and sex-specific mortality patterns, we focused on the 10 leading causes of death based on the 2022 data from Statistics Korea.

For males, the top 10 causes of death included malignant neoplasms (C00–C97), heart disease (I20–I51), COVID-19 (U07.1, U07.2, U10), pneumonia (J12–J18), cerebrovascular disease (I60–I69), intentional self-harm (suicide; X60–X84), diabetes (E10–E14), liver disease (K70–K76), chronic respiratory diseases (J40–J47), and Alzheimer’s disease (G30). Meanwhile, for females, the top 10 causes of death included malignant neoplasms (C00–C97), heart disease (I20–I51), COVID-19 (U07.1, U07.2, U10), cerebrovascular disease (I60–I69), pneumonia (J12–J18), Alzheimer’s disease (G30), diabetes (E10–E14), hypertensive diseases (I10–I13), blood poisoning (A40–41), and deliberate self-harm (suicide; X60–X84).

### 2.2. Decomposing Life Expectancy Changes

To examine the interval-by-interval trends in life expectancy changes in Korea, the Joinpoint Regression Program (version 5.2.0) of the National Cancer Institute (NCI) was used. This analysis employed the Annual Percent Change (APC) as a key metric [24,25,26].

In this study, the number of join points was 2 through the Monte Carlo Permutation method, and the APC and Average Annual Percent Change (AAPC) were calculated. The two-sided test estimates the confidence interval for the slope of the regression line, with statistical significance defined as a two-sided *p*-value < 0.05.

To further understand the contributions of age, sex, and causes of death to life expectancy changes in Korea, we applied the Arriaga decomposition method (Arriaga, 1984; Chisumpa and Odimegwu, 2018) [27].

The change in life expectancy in each age group is defined as the sum of the direct (DE), indirect (IE), and interaction effects (I) due to the change in mortality rate in that age group. DE and IE exist independently of each other and are affected by the change in mortality rate within the age group. The IE shows how the number of survivors at age n + i after that age changes from the change in mortality rate in a specific age range. Since mortality rates change simultaneously in all ages, survivors at a specific age are exposed to various levels of mortality, which can be explained by the interaction effect. DE refers to the impact of the change in mortality rate within a specific age group on life expectancy for that same age group. It is calculated by multiplying the ratio of the number of survivors at age *a* at time t to survivors at age *x* at time t with the ratio of survivors at age *x* at time t, after accounting for the difference in life expectancy (ext+ni−exti) of a specific age group (*x*, *x* + *i*) at two times (*t*, *t* + *n*).DExi=ltxlat(ext+ni−exti)

IE represents the effect of a decrease in mortality within a particular age group on life expectancy by changing the number of survivors (*CS*) at the end of that age group. This can be calculated using the following formula.IExi=CSxilatex+it=Tx+itlatlxtlx+it+nlx+itlt+nx−1

*I* refers to the effect that occurs when survivors of a certain age group are exposed to a new mortality level when they reach the age of that age group or higher. This is due to the change in mortality at all ages, and thus the effect of years of survival (*OE*) must also be considered as additional survivors are exposed to the new mortality rate. The difference between the two is defined as the IE.OExi=CSxilatex+it+n=Tx+it+nlatlxtlxt+n−lx+itlx+1tIxi=OExi−IExi

The change in life expectancy is due to the change in mortality at each specific age and can be viewed as an increase or decrease in the number of survival years due to the change in the mortality rate (total central mortality rate) on the life table. The contribution years to the increase in life expectancy due to the change in mortality in a specific age group from the base year (year 0) to the fertilizer year (year 1), that is, *SAC_j_*(*_i_e_x_*) (change mortality in a specific age group, SAC), can be expressed by the following equation.SACj(exi)=F(mj1−mj0) 

If the change in total mortality in age group j is represented by $C_{j}=m_{j}^{1}-m_{j}^{0}$, the change in mortality for each cause (*C*) can be represented by Ccj=mj1c−mj0c. The change in mortality in a specific age group Cj=∑c=1sCjc (*S* is the total mortality group) is equal to the sum of the changes in mortality by each cause in the same age group, and the contribution years to life expectancy created by the change in each cause in a specific age group can be represented as a ratio to total mortality.SACjc(exi)=SACj(exi)CjcCj

## 3. Results

### 3.1. Life Expectancy Trends in Korea

We used the Joinpoint regression analysis to examine the trends in life expectancy over the past 30 years. From 1990 to 2022, the life expectancy in Korea showed significant join points in 2007 and 2020.

Male life expectancy increased significantly by an average of 0.70 years annually (95% CI: 0.68–0.72) from 1990 to 2007. However, the annual increase slowed down to 0.47 years per year (95% CI: 0.45–0.49) after 2007, reaching 80.49 years in 2020. In 2020, during the COVID-19 pandemic, life expectancy began to decline (−0.36 years per year, 95% CI: −0.68–−0.03), dropping to 79.86 years in 2022.

The trend in life expectancy for women was similar to that for men. From 1990 to 2007, it increased by an average of 0.48 per year (95% CI: 0.47–0.50). This rate slowed to 0.34 per year (95% CI: 0.3–0.4) after 2007, reaching 86.47 years in 2020. In 2020, during the COVID-19 pandemic, life expectancy for women began to decline (−0.45 per year, 95% CI: −0.71 to −0.18), reaching 85.62 years in 2022 (Figure 1). Our Joinpoint analysis revealed that significant trend changes in 2020 coincide with the start of the COVID-19 pandemic in 2020.

### 3.2. Contributions to Life Expectancy by Age and Sex

We identified a significant join point in life expectancy in 2020, coinciding with the onset of the COVID-19 pandemic. To analyze the contributions of age, sex, and cause of death to changes in life expectancy between 2019 (pre-pandemic) and 2022 (post-pandemic) we used the Arriaga decomposition method.

Age-specific factors contributed to the decline in life expectancy across most age groups (Table 1, Figure 2). For men, the largest decreases in life expectancy occurred in the following order: 0.129 years (31.30%) for ages 85–89 years, 0.105 years (25.44%) for ages 80–84 years, 0.083 years (20.25%) for ages 90–94 years, and 0.068 years (−16.46%) for ages 70–74 years.

Conversely, it contributed to an increase in life expectancy in the following order: 0.034 years (8.25%) for ages 50–54 years, 0.023 years (5.55%) for ages 45–49 years, and 0.020 years (4.78%) for ages 55–59 years.

However, for women, the contribution to the increase in life expectancy was minimal, with a decrease in life expectancy in the following order: 0.177 years (25.97%) for ages 85–89 years, 0.140 years (20.56%) at ages 90–94 years, and 0.135 years (19.76%) for ages 80–84 years. Additionally, the life expectancy of women decreased by 0.268 years more than that of men.

The overall decline in life expectancy in 2022 compared with that in 2019 is analyzed to have been contributed by the older population group aged ≥ 80 years.

### 3.3. Contributions to Life Expectancy by Top 10 Causes of Death

This study analyzed the contributions of major causes of death to changes in life expectancy in Korea using life tables and cause-of-death statistics for 2019 and 2022.

When analyzing the contributions to male life expectancy by these causes, malignant neoplasms (0.299 years), heart disease (0.053 years), intentional self-harm (suicide; 0.35 years), chronic respiratory diseases (0.035 years), and pneumonia (0.019 years) contributed to the increase in life expectancy in that order. COVID-19, however, contributed the most to the decrease in male life expectancy (−0.605 years), surpassing the combined decreases from other causes (−0.412 years) from 2019 to 2022 (Table 2).

Further age-specific analyses revealed that malignant neoplasms contributed the most to increased life expectancy among individuals aged 55–54 years (0.045 years), followed by 65–69 years (0.043 years) and 80–84 years (0.042 years). Kidney disease contributed 0.034 years to the increase in life expectancy among individuals aged ≥ 65 years. Intentional self-harm (suicide) increased life expectancy by 0.074 years among individuals aged ≥ 35 years. Chronic respiratory diseases and pneumonia contributed 0.002 years to life expectancy among those aged ≥ 80 years, while their effects in other age groups were minimal.

COVID-19 contributed substantially to the decline in life expectancy. The greatest losses were observed in 80–84 years (−0.125 years), followed by 85–89 years (−0.097 years), and ≥90 years (−0.084 years) old.

Other notable findings revealed that Alzheimer’s disease contributed to a decline in life expectancy in individuals aged ≥ 80 years (−0.026 years), as well as diabetes in those aged 80–84 years (−0.006 years) and 60–64 years (−0.006 years). Cerebrovascular and liver diseases contributed to the largest decrease in life expectancy for individuals aged 85–89 (−0.005 years) and 40–44 (−0.007 years), respectively (Table 2, Figure 3).

When analyzing the contribution of the top 10 causes of death to female life expectancy, malignant neoplasms contributed 0.110 years, followed by heart disease (0.067 years) and pneumonia (0.017 years), leading to an increase in life expectancy in that order. Conversely, COVID-19 (−0.621 years) had the most substantial negative impact on life expectancy, followed by Alzheimer’s disease (−0.072 years), diabetes (−0.032 years), sepsis (−0.021 years), cerebrovascular disease (−0.018 years), and hypertensive diseases (−0.010 years).

Malignant neoplasms were the primary contributors to the increase in life expectancy among women, with the most remarkable contribution observed in the 55–59 years age group (0.020 years). However, this condition contributed to increased life expectancy in individuals aged ≥ 25 years while slightly decreasing life expectancy in younger age groups. For kidney disease and pneumonia, the largest contributions to increased life expectancy were observed in those aged 85–89 years, accounting for 0.017 and 0.012 years, respectively.

COVID-19 was a major contributor to the decline in female life expectancy across all age groups. The most considerable impact was observed in the 85–89 years age group (−0.142 years), followed by 80–84 years (−0.137 years) and >90 years (−0.129 years) old age group.

Alzheimer’s disease was a major cause of decreased life expectancy in women aged ≥ 80 years (−0.062 years). Additionally, diabetes, sepsis, hypertensive diseases, and cerebrovascular disease were the leading causes of death in those aged ≥ 80 years, in that order (Table 3, Figure 3).

## 4. Discussion

Economic development and advancements in medical technology have markedly increased the life expectancy in Korea. Since 1956, life expectancy in Korea has exceeded the global average [5]. Our analysis suggests that since 1990, the age group of ≥65 years has substantially contributed to this increase. Recently, contributions to life expectancy from younger age groups, including neonates, have gradually decreased, while contributions from middle-aged and older population groups have increased [2,8].

This trend of increasing life expectancy in Korea has slowed somewhat due to the COVID-19 pandemic. To investigate this, our study analyzed relevant trends. Using Joinpoint regression analysis [24,25,26], we identified two significant join points in 2007 and 2020. In 2007, the pace of life expectancy growth slowed, although it remained positive. However, in 2020, coinciding with the onset of the COVID-19 pandemic, life expectancy began to decline, with increases recorded for both men (−0.36 years per year, 95% CI: −0.68 to −0.03) and women (−0.45 years per year, 95% CI: −0.71 to −0.18). We used Joinpoint regression analysis to identify a significant join point in life expectancy in South Korea in 2020, coinciding with the start of the COVID-19 pandemic.

Based on these findings, further analysis was conducted using the Arriaga decomposition method [27,28] to examine the contributions by age, sex, and cause of death in 2019 and 2022, representing periods before and after the COVID-19 pandemic. Results showed that declines in life expectancy occurred across most age groups, with the most substantial decreases observed in individuals aged ≥ 80 years (male: −0.35 years; female: −0.52 years). Women contributed a more significant decline in life expectancy (−0.68 years) than men (−0.41 years).

COVID-19 emerged as a major contributor to reduced life expectancy among the causes of death, particularly among individuals aged ≥ 80 years (men: −0.306 years; women: −0.408 years). The decline in life expectancy during the pandemic is primarily attributed to deaths among older individuals due to COVID-19. Studies in other Asian countries have similarly shown that the COVID-19 poses a greater threat to older individuals and men, whose immune systems are biologically more vulnerable [6,29]. Additionally, a study on changes in life expectancy due to the COVID-19 pandemic in 29 countries, including the United States and Europe, reported a widening sex-specific gap in life expectancy due to an increase in the life expectancy of women [2].

However, this study revealed that in Korea, COVID-19-related deaths among older individuals affected more women than men, narrowing the sex-specific gap in life expectancy by 0.3 years. A 2022 report on social trends in Korea also indicated that the COVID-19 mortality rate for women was slightly higher than for men. This difference was attributed to the disproportionate sex distribution in the older adult population aged ≥ 80, where COVID-19 deaths are concentrated, with women comprising 66.4% and men 33.6% [30].

Globally, while Korea was relatively less affected by the pandemic compared to other nations, some notable patterns were observed in the Joinpoint regression analysis results. According to Kao et al. [4], among 107 countries and territories (excluding Africa and Oceania), 45.3% experienced a significant decline in life expectancy during the most recent period, and 77 countries and territories (32.6%) showed significant declines between 2019 and 2021 [4].

Before the COVID-19 pandemic, the ten leading causes of death in Korea included malignant neoplasms, heart disease, and cerebrovascular diseases, most of which are non-infectious diseases. However, following the COVID-19 pandemic, life expectancy was expected to decline (−0.573 years for males), with a major impact on the −0.55-year period.

In contrast, a change in the contribution pattern of specific causes of death to life expectancy pre- and post-pandemic was observed. Analyses indicated that intentional self-harm (suicide) contributed to increased life expectancy after the pandemic.

According to the GBD 2021 Disease and Injury Burden Analysis, global life expectancy has decreased, with COVID-19 becoming the leading cause of death worldwide. Despite this, significant health improvements have been reported for other non-communicable and infectious diseases [31].

A study by Kuehn BM et al. compared actual 2020 mortality data with the estimated 2020 life expectancy based on trends from 2005 to 2019 for 37 high-income countries. The findings revealed that Russia, the United States, and Bulgaria experienced the largest declines in life expectancy. In contrast, no changes were observed in Denmark, Iceland, and Korea, while New Zealand, Taiwan, and Norway saw slight increases. In the 31 countries where life expectancy declined, an estimated 28 million additional years of life were lost in 2020, five times the excess deaths associated with the seasonal influenza pandemic of 2015 [12,32,33,34].

Based on the findings reported by Kao et al., the COVID-19 pandemic caused an estimated 14.9 million excess deaths worldwide in 2020 and 2021. These excess deaths are considered the primary reason for the decline in life expectancy observed in many countries [4].

South Korea is widely known for its COVID-19 pandemic response, employing a preemptive testing, tracing, and treatment (3T) strategy. As a result, life expectancy in the country remained relatively unchanged. This study examines the impact of the COVID-19 pandemic on life expectancy in Korea using Joinpoint regression analysis and the Arriaga decomposition method. It serves as a basic resource for health care initiatives aimed at preparing for future pandemics. In addition, it proposes a health management system for older adults as well as policies for preventing and managing chronic diseases, strengthening public health care systems and infectious disease expertise, and establishing collaborative systems with private medical institutions for future infectious disease response.

In a study on the impact of COVID-19 on life expectancy, the Korean data extended up to 2022, whereas global data were limited to 2021. Owing to data limitations, our study could not cover the most recent trends. Future research should consider the potential relationship between excess deaths from COVID-19 and life expectancy. Employing advanced methodologies, such as the Lee–Carter model, could provide valuable insights. Moreover, comprehensive analyses leveraging AI and other technologies are required, integrating data beyond simple life tables and cause-of-death statistics to include the indirect impacts of the pandemic, such as the effects of vaccines, healthcare systems, and health policies.

## Figures and Tables

**Figure 1 healthcare-13-00258-f001:**
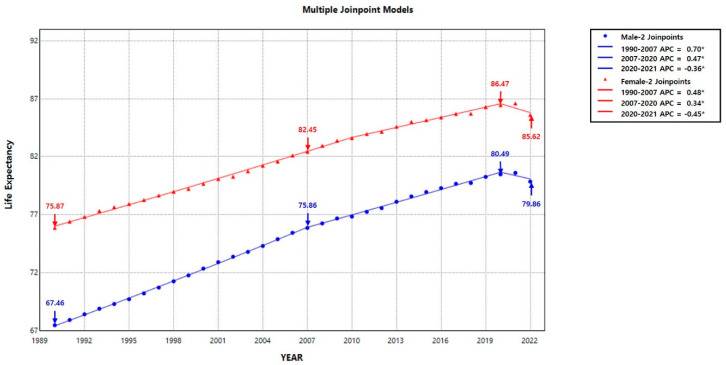
Temporal patterns in life expectancy in Korea from 1990 to 2022. The annual percentage change (APC) is significantly different from zero at the alpha = 0.05 level, as indicated by an asterisk (*). Blue arrows represent life expectancy for males in key years (1990, 2007, 2020, and 2022), and red arrows represent that for females.

**Figure 2 healthcare-13-00258-f002:**
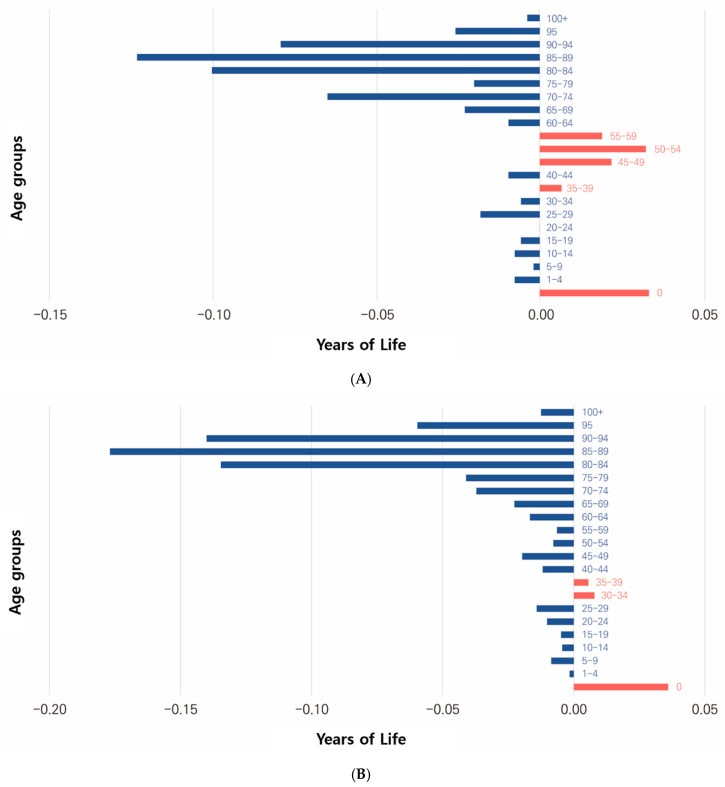
Mortality contributions to life expectancy changes by age group, 2019 and 2020. (**A**) Contributions of different age groups to male life expectancy changes between 2019 and 2020. (**B**) Contributions of different age groups to female life expectancy changes between 2019 and 2020.

**Figure 3 healthcare-13-00258-f003:**
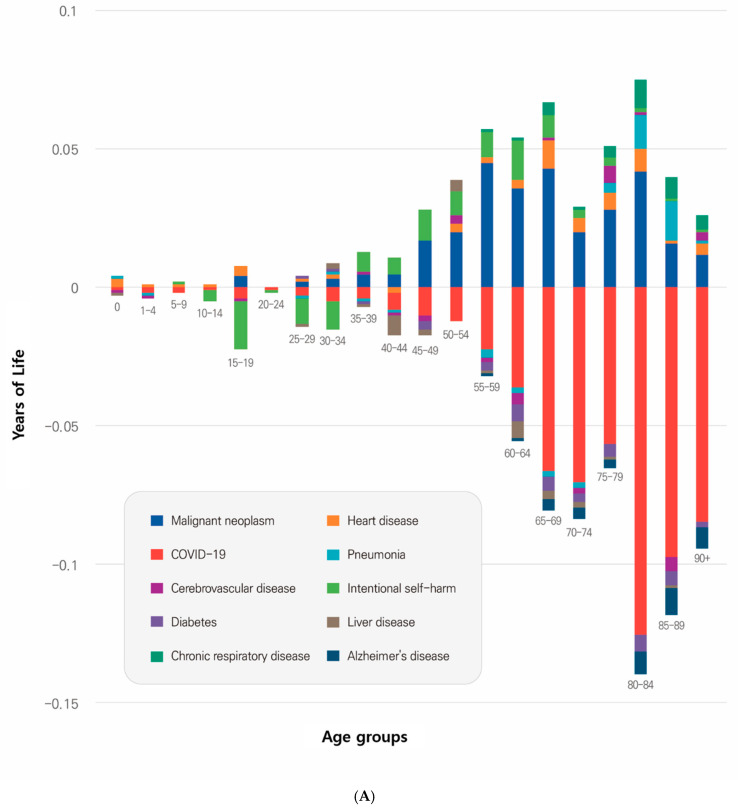
Contributions of the top 10 causes of death to life expectancy changes, 2019–2022. (**A**) Analysis of life expectancy changes for males due to the top 10 causes of death from 2019 to 2022. (**B**) Analysis of life expectancy changes for females due to the top 10 causes of death from 2019 to 2022.

**Table 1 healthcare-13-00258-t001:** Contributions to life expectancy changes by age group in Korea, 2019–2022.

Age	Male		Female		A-B
	Year (A)	%	Year (B)	%	
0	0.035	8.51	0.036	5.29	−0.001
1–4	−0.008	−1.89	−0.002	−0.23	−0.006
5–9	−0.002	−0.51	−0.009	−1.26	0.007
10–14	−0.008	−1.96	−0.004	−0.65	−0.004
15–19	−0.006	−1.41	−0.005	−0.71	−0.001
20–24	0.000	−0.08	−0.010	−1.49	0.010
25–29	−0.019	−4.58	−0.014	−2.07	−0.005
30–34	−0.006	−1.49	0.008	1.16	−0.014
35–39	0.007	1.68	0.006	0.82	0.001
40–44	−0.010	−2.48	−0.012	−1.74	0.002
45–49	0.023	5.55	−0.020	−2.89	0.043
50–54	0.034	8.25	−0.008	−1.14	0.042
55–59	0.020	4.78	−0.006	−0.94	0.026
60–64	−0.010	−2.49	−0.017	−2.46	0.007
65–69	−0.024	−5.80	−0.023	−3.32	−0.001
70–74	−0.068	−16.46	−0.037	−5.46	−0.031
75–79	−0.021	−4.99	−0.041	−6.03	0.020
80–84	−0.105	−25.44	−0.135	−19.76	0.030
85–89	−0.129	−31.30	−0.177	−25.97	0.048
90–94	−0.083	−20.25	−0.140	−20.56	0.057
95–99	−0.027	−6.64	−0.060	−8.76	0.033
100+	−0.004	−1.01	−0.012	−1.83	0.008
**Total**	−0.412	−100.00	−0.681	−100.00	0.269

Legend: This table provides an overview of age-specific contributions to life expectancy changes by sex. Negative values indicate declines in life expectancy, while positive values reflect increases.

**Table 2 healthcare-13-00258-t002:** Contributions of the top 10 causes of death to male life expectancy changes (2019–2022).

Male	Malignant Neoplasms	Heart Diseases	COVID-19	Pneumonia	Cerebrovascular Diseases	Intentional Self-Harm (Suicide)	Diabetes	Liver Disease	Chronic Respiratory Disease	Alzheimer’s Disease	Other
0	0.000	0.003	−0.001	0.001	−0.001	0.000	0.000	−0.001	0.000	0.000	0.034
1–4	0.000	0.001	−0.002	−0.001	−0.001	0.000	0.000	0.000	0.000	0.000	−0.005
5–9	0.000	0.001	−0.002	0.000	0.000	0.001	0.000	0.000	0.000	0.000	−0.002
10–14	0.000	0.001	−0.001	0.000	0.000	−0.004	0.000	0.000	0.000	0.000	−0.004
5–19	0.004	0.004	−0.004	0.000	−0.001	−0.017	0.000	0.000	0.000	0.000	0.007
20–24	0.000	0.000	−0.001	0.000	0.000	−0.001	0.000	0.000	0.000	0.000	0.001
25–29	0.002	0.001	−0.003	−0.001	0.000	−0.009	0.001	−0.001	0.000	0.000	−0.009
30–34	0.003	0.002	−0.005	0.001	0.000	−0.010	0.001	0.002	0.000	0.000	0.000
35–39	0.005	0.000	−0.004	−0.001	0.001	0.007	−0.001	−0.001	0.000	0.000	0.002
40–44	0.005	−0.002	−0.006	−0.001	−0.001	0.006	0.000	−0.007	0.000	0.000	−0.004
45–49	0.017	0.000	−0.010	0.000	−0.002	0.011	−0.003	−0.002	0.000	0.000	0.010
50–54	0.020	0.003	−0.012	0.000	0.003	0.009	0.000	0.004	0.000	0.000	0.007
55–59	0.045	0.002	−0.022	−0.003	−0.002	0.009	−0.003	−0.001	0.001	−0.001	−0.004
60–64	0.036	0.003	−0.036	−0.002	−0.004	0.014	−0.006	−0.006	0.001	−0.001	−0.010
65–69	0.043	0.010	−0.066	−0.002	0.001	0.008	−0.005	−0.003	0.005	−0.004	−0.011
70–74	0.020	0.005	−0.070	−0.002	−0.002	0.003	−0.003	−0.002	0.001	−0.004	−0.016
75–79	0.028	0.006	−0.056	0.004	0.006	0.003	−0.005	−0.001	0.004	−0.003	−0.007
80–84	0.042	0.008	−0.125	0.012	0.001	0.002	−0.006	0.000	0.010	−0.008	−0.040
85–89	0.016	0.001	−0.097	0.014	−0.005	0.001	−0.005	−0.001	0.008	−0.010	−0.317
90+	0.012	0.004	−0.084	0.001	0.003	0.001	−0.002	0.000	0.005	−0.008	−0.047
**Total**	0.299	0.053	−0.605	0.019	−0.003	0.035	−0.038	−0.020	0.035	−0.040	−0.414

**Table 3 healthcare-13-00258-t003:** Contributions of the top 10 causes of death for female life expectancy changes (2019–2022).

Female	Malignant Neoplasms	Heart Diseases	COVID-19	Cerebrovascular Diseases	Pneumonia	Alzheimer’s Disease	Diabetes	Hypertensive Disease	Sepsis	Intentional Self-Harm (Suicide)	Others
0	0.000	0.002	−0.002	−0.002	0.001	0.000	0.000	0.000	−0.001	0.000	0.037
1–4	−0.001	0.000	−0.001	0.000	−0.001	0.000	0.000	0.000	0.000	0.000	0.002
5–9	−0.005	0.000	−0.007	−0.002	−0.003	0.000	0.000	0.000	0.002	0.000	0.008
10–14	−0.001	0.000	−0.001	−0.001	0.000	0.000	0.000	0.000	0.000	−0.003	0.000
15–19	0.000	0.002	−0.002	0.001	0.000	0.000	0.000	0.000	0.000	0.000	−0.005
20–24	−0.001	0.000	−0.001	−0.001	0.000	0.000	0.000	0.000	0.000	−0.002	−0.005
25–29	0.001	−0.001	−0.002	−0.002	0.000	0.000	−0.001	0.000	0.000	−0.006	−0.004
30–34	0.000	0.001	0.002	0.000	−0.001	0.000	0.000	0.000	0.001	0.000	0.001
35–39	0.002	0.000	−0.001	0.000	0.000	0.000	0.000	0.000	0.000	0.003	−0.001
40–44	0.003	−0.001	−0.006	−0.001	0.000	0.000	0.000	0.000	0.001	−0.005	−0.003
45–49	0.001	0.000	−0.007	−0.001	−0.001	0.000	−0.001	0.000	−0.001	0.001	−0.011
50–54	0.005	0.001	−0.008	0.002	0.000	0.000	−0.001	0.000	0.001	−0.001	−0.006
55–59	0.020	−0.002	−0.018	−0.001	−0.002	0.000	−0.003	−0.001	−0.001	0.004	−0.003
60–64	0.012	0.003	−0.021	−0.001	−0.001	−0.001	−0.002	−0.001	−0.002	0.002	−0.006
65–69	0.014	0.004	−0.036	−0.001	0.000	−0.002	0.000	0.000	−0.001	0.000	0.000
70–74	0.008	0.005	−0.042	−0.002	−0.001	−0.003	−0.002	0.001	−0.002	0.003	−0.002
75–79	0.016	0.014	−0.061	0.000	0.002	−0.005	−0.004	0.002	−0.003	0.002	−0.004
80–84	0.014	0.016	−0.137	0.000	0.007	−0.014	−0.004	−0.001	−0.003	0.002	−0.015
85–89	0.014	0.017	−0.142	−0.006	0.012	−0.021	−0.010	−0.001	−0.005	0.001	−0.448
90+	0.007	0.006	−0.129	−0.001	0.005	−0.027	−0.006	−0.010	−0.004	0.000	−0.054
**Total**	0.110	0.067	−0.621	−0.018	0.017	−0.072	−0.032	−0.010	−0.021	0.001	−0.518

## Data Availability

The raw data supporting the conclusions of this article will be made available by the authors on request.

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
