# Peer review of "Impact of the COVID-19 Pandemic on Life Expectancy in South Korea, 2019–2022"

_healthcare, 2025, doi:10.3390/healthcare13030258_

Round 1
Reviewer 1 Report
Comments and Suggestions for Authors
General comments.
The aim of the study is to analyse changes in life expectancy in South Korea due to COVID-19 with special attention to the impact of sex, age, and selected main causes of death. The results may raise interest of international readers as Korea has had one of the longest life expectancies in the world and mortality due to COVID-19 was relatively low. The data source was official South Korea and changes in life expectancy were analysed using the appropriate methodology by jointpoint regression analysis and stratification using the Arriaga decomposition method. The overall quality of the article is good, but some comments for minor additions will follow.
Specific comments.
Introduction.
1. Additional information about novelty of the study and its results could be added compared to previous life expectancy studies in South Korea (e.g. https://www.frontiersin.org/journals/public-health/articles/10.3389/fpubh.2023.1215914/full )
Methods.
1. For Section 2.1. few words about the completeness and quality of the demographic data of South Korea can be added for the international reader.
2. Despite the presence of a reference to the source where the Arriaga decomposition method is described, it is probably worse to add short information about the essence of the method. Most of those who do not regularly work with life expectancy analysis are probably unfamiliar with it.
Results.
1. Figure 1. For the box with APC, no need to put the decimal number in years.
2. Figure 2. The letters A and B are omitted in the title of the figure.
3. Lines 143 – 155 fit in the “Materials and methods” part.
4. Tables 2 and 3. In headers of columns, the names of causal groups instead of ICD codes would be preferable to increase the readability of the tables.
5. Figure 3. The same as for Figure 2.
Discussion.
1. Line 226. Please, add interpretation of this jointpoint. Is it a statistical phenomenon (just reaching statistical significance) or is there a period effect for South Korea?
2. Lines 235 – 249. What could be the explanation for the narrowing of the gap in the sex groups and the increase in COVID-19 specific mortality in elderly women in South Korea? (Additionally, in the scientific research literature, the term “sex” would be preferable instead to the social construct of the “gender”.)
3. Those people who died by the primary cause of COVID-19 may have had other diseases from this study before the infection, and mortality from these diseases (impact on specific changes in life expectancy) may be related to death by COVID-19. Can changes in life expectancy by age / causal groups be interpreted in the context of COVID-19 infection and death?
Author Response
Comments 1) Introduction. 1. Additional information about novelty of the study and its results could be added compared to previous life expectancy studies in South Korea (e.g. https://www.frontiersin.org/journals/public-health/articles/10.3389/fpubh.2023.1215914/full )
Response 1) Thank you for pointing this out. We agree with this comment. Therefore, we have revised and supplemented lines 82-86 on page 2.
Comments 2) Methods. 1. For Section 2.1. few words about the completeness and quality of the demographic data of South Korea can be added for the international reader.
Response 2) Thank you for pointing this out. We agree with this comment. Therefore, We have revised and supplemented lines 93~94 on page 3.
Comments 3) Methods. 2. Despite the presence of a reference to the source where the Arriaga decomposition method is described, it is probably worse to add short information about the essence of the method. Most of those who do not regularly work with life expectancy analysis are probably unfamiliar with it.
Response 3) Thank you for pointing this out. We agree with this comment. Therefore, We have revised and supplemented lines 117~154 on page 3~4.
Comments 4) Results. 1. Figure 1. For the box with APC, no need to put the decimal number in years.
Response 4) Thank you for pointing this out. We agree with this comment. Therefore, We have revised and supplemented Figure 1 on page 5.
Comments 5) Results. 2. Figure 2. The letters A and B are omitted in the title of the figure.
Response 5) Thank you for pointing this out. We agree with this comment. Therefore, We have revised and supplemented Figure 2 on page 7.
Comments 6) Results. 3. Lines 143 – 155 fit in the “Materials and methods” part.
Response 6) Thank you for pointing this out. We agree with this comment. Therefore, We have revised and supplemented lines 95~105 on page 3.
Comments 7) Results. 4. Tables 2 and 3. In headers of columns, the names of causal groups instead of ICD codes would be preferable to increase the readability of the tables.
Response 7) Thank you for pointing this out. We agree with this comment. Therefore, We have revised and supplemented Table 2&Table 3 on page 8~11.
Comments 8) Results. 5. Figure 3. The same as for Figure 2.
Response 8) Thank you for pointing this out. We agree with this comment. Therefore, We have revised and supplemented Figure 3 on page 12.
Comments 9) Discussion. 1. Line 226. Please, add interpretation of this jointpoint. Is it a statistical phenomenon (just reaching statistical significance) or is there a period effect for South Korea?
Response 9) Thank you for pointing this out. We agree with this comment. Therefore, We have revised and supplemented lines 281-283 on page 13.
Comments 10) Discussion. 2. Lines 235 – 249. What could be the explanation for the narrowing of the gap in the sex groups and the increase in COVID-19 specific mortality in elderly women in South Korea? (Additionally, in the scientific research literature, the term “sex” would be preferable instead to the social construct of the “gender”.)
Response 10) Thank you for pointing this out. We agree with this comment. Therefore, We have revised and supplemented lines 302~306 on page 13.
Comments 11) Discussion. 3. Those people who died by the primary cause of COVID-19 may have had other diseases from this study before the infection, and mortality from these diseases (impact on specific changes in life expectancy) may be related to death by COVID-19. Can changes in life expectancy by age / causal groups be interpreted in the context of COVID-19 infection and death?
Response 11) Thank you for pointing this out. We agree with this comment. But, It is difficult to interpret the changes in life expectancy by age/cause of death in terms of the relationship between COVID-19 infection and pre-existing diseases. There is currently no research on the part you suggested, and I think future research is needed.
Reviewer 2 Report
Comments and Suggestions for Authors<A brief summary >
This research investigated the change points of life expectancy and showd the impacts of Covid19 pandemic in decreasing life expectancy in Korea through the Joinpoint regression and Arriaga decomposition methods.
<General concept comments>
Authors should provide a more detailed explanation of the Joinpoint regression and Arriaga decomposition to ensure readers can fully understand the methodology. While the proposed methods are useful for capturing overall trends, they appear to rely solely on a simple life table and do not account for various factors that could contribute to mortality. In the Discussion section, an integrated methodology using models such as the Lee-Carter model is proposed as a future direction. However, it is crucial to first include a detailed explanation of the current methodology and its limitations in the main text.
<Specific comments>
1. As a result of Joinpoint regression analysis, the authors identified two change points, 2007 and 2020. The considerate explanation or assumption should be added on 2007 change point, as well as 2020.
2. In line 105, “82.45” should be corrected to “86.47”.
3. In reviewing Table 1, I suggest making it more refined and precise to enhance clarity and readability. Specifically:
1) Column Headings: Ensure the column names are labeled with utmost accuracy and specificity. Use clear and concise terminology to avoid any ambiguity.
2) Units of Measurement: Explicitly indicate the units of measurement for each numerical value in the table. This will help readers understand the context of the data without referring to other parts of the manuscript.
4. In reviewing Table 4, for author-friendly policy, please add specific disease name as column title directly.
Author Response
Comments 1) Authors should provide a more detailed explanation of the Joinpoint regression and Arriaga decomposition to ensure readers can fully understand the methodology. While the proposed methods are useful for capturing overall trends, they appear to rely solely on a simple life table and do not account for various factors that could contribute to mortality. In the Discussion section, an integrated methodology using models such as the Lee-Carter model is proposed as a future direction. However, it is crucial to first include a detailed explanation of the current methodology and its limitations in the main text.
Response 1) Thank you for pointing this out. We agree with this comment. Therefore, we have revised and supplemented lines 110~155 on page 3~4.
Comments 2) As a result of Joinpoint regression analysis, the authors identified two change points, 2007 and 2020. The considerate explanation or assumption should be added on 2007 change point, as well as 2020.
Response 2) Thank you for pointing this out. We agree with this comment. Therefore, we have revised and supplemented lines 170~172 on page 4.
Comments 3) In line 105, “82.45” should be corrected to “86.47”.
Response 3) Thank you for pointing this out. We agree with this comment. Therefore, we have revised and supplemented lines 168 on page 4.
Comments 4) In reviewing Table 1, I suggest making it more refined and precise to enhance clarity and readability. Specifically:
1) Column Headings: Ensure the column names are labeled with utmost accuracy and specificity. Use clear and concise terminology to avoid any ambiguity.
2) Units of Measurement: Explicitly indicate the units of measurement for each numerical value in the table. This will help readers understand the context of the data without referring to other parts of the manuscript.
Response 4) Thank you for pointing this out. We agree with this comment. Therefore, we have revised and supplemented Table1 on page 5.
Comments 5) In reviewing Table 4, for author-friendly policy, please add specific disease name as column title directly.
Response 5) Thank you for pointing this out. We agree with this comment. Therefore, we have revised and supplemented Table 2&Table 3 on page 8~11.
Reviewer 3 Report
Comments and Suggestions for Authors
1. Introduction: Although the impact of COVID-19 is mentioned, the innovation of this study compared to previous research is not clearly stated.
2. Methods: The data source is mentioned, but the specific data processing steps are not clarified.
3. Discussion: While data from multiple countries is cited, the uniqueness of South Korea's specific preventive measures is not sufficiently highlighted or compared. The practical significance of the research findings remains underexplored, particularly regarding their implications for policy and society. Furthermore, the study's limitations and potential directions for future research are not adequately addressed.
4. Table Format: Tables should be adjusted to the three-line table format.
Author Response
Comments 1) Introduction: Although the impact of COVID-19 is mentioned, the innovation of this study compared to previous research is not clearly stated.
Response 1) Thank you for pointing this out. We agree with this comment. Therefore, we have revised and supplemented lines 82-86 on page 2.
Comments 2) Methods: The data source is mentioned, but the specific data processing steps are not clarified.
Response 2) Thank you for pointing this out. We agree with this comment. Therefore, we have revised and supplemented lines 110~155 on page 3.
Comments 3) Discussion: While data from multiple countries is cited, the uniqueness of South Korea's specific preventive measures is not sufficiently highlighted or compared. The practical significance of the research findings remains underexplored, particularly regarding their implications for policy and society. Furthermore, the study's limitations and potential directions for future research are not adequately addressed.
Response 3) Thank you for pointing this out. We agree with this comment. Therefore, we have revised and supplemented it in lines 336-345.
Comments 4) Table Format: Tables should be adjusted to the three-line table format.
Response 4) Thank you for pointing this out. We agree with this comment. Therefore, we have revised and supplemented Table 1, Table 2 & Table 3 on page 5~11.